# Future phytoplankton diversity in a changing climate

Stephanie A. Henson [1✉], B. B. Cael[1], Stephanie R. Allen[1,2,5] & Stephanie Dutkiewicz [3,4]

The future response of marine ecosystem diversity to continued anthropogenic forcing is poorly constrained. Phytoplankton are a diverse set of organisms that form the base of the marine ecosystem. Currently, ocean biogeochemistry and ecosystem models used for climate change projections typically include only 2−3 phytoplankton types and are, therefore, too simple to adequately assess the potential for changes in plankton community structure. Here, we analyse a complex ecosystem model with 35 phytoplankton types to evaluate the changes in phytoplankton community composition, turnover and size structure over the 21st century. We find that the rate of turnover in the phytoplankton community becomes faster during this century, that is, the community structure becomes increasingly unstable in response to climate change. Combined with alterations to phytoplankton diversity, our results imply a loss of ecological resilience with likely knock-on effects on the productivity and functioning of the marine environment.

[1] National Oceanography Centre, European Way, Southampton, UK. [2] School of Ocean and Earth Sciences, University of Southampton, Waterfront Campus, European Way, Southampton, UK. [3] Center for Global Change Science, Massachusetts Institute of Technology, Cambridge, MA, USA. [4] Earth, Atmospheric and Planetary Sciences, Massachusetts Institute of Technology, Cambridge, MA, USA. [5] Present address: Plymouth Marine Laboratory, Prospect Place, Plymouth, UK. ✉email: s.henson@noc.ac.uk

The socio-economic services provided by marine ecosystems are critical to human wellbeing. For example, fisheries provide almost half of Earth's population with at least 20% of their animal protein intake[1]. Marine ecosystems also regulate Earth's climate by absorbing and sequestering atmospheric $CO_2$. Therefore, maintaining biodiversity is critical to providing resilience against future climate change and extremes[2]. At a global scale, biodiversity loss is being driven by human activities[3,4], although clear trends of biodiversity decline in local ecosystems have proven difficult to identify[5–7]. Rather, the dominant species appear to be rapidly turned over, resulting in widespread reorganisation of ecosystems. These changes are potentially even more pronounced in the oceans than in the terrestrial realm[8].

In addition to human pressures on habitat, anthropogenic climate change is likely to drive biodiversity loss and hence decrease ecosystem stability[2,9], thus affecting both the functioning and structure of marine ecosystems[10–12]. Ocean warming and alterations to nutrient supply via changing circulation or stratification, combined with additional stressors such as ocean acidification and deoxygenation, are likely to force community reorganisation. Predicting future changes to marine ecosystems is challenging, partly due to the relative paucity of consistent, repeated sampling, the inherent variability over daily to inter-annual scales in community composition[13,14], and the lack of knowledge of how future climate change and other anthropogenic stressors may combine to alter biodiversity[15]. However, with future oceans predicted to be ~ 2−4 °C warmer, more acidic, and reduced in oxygen concentration[16], species must adapt, migrate to regions of analogous conditions, or face extinction[17–19]. The expected resulting changes to biodiversity are likely to affect fundamental ecosystem functioning and processes, such as biomass production and maintaining water quality[20–22], as well as the entire marine ecosystem structure, with consequences for the ocean's capacity for food production and climate regulation[23].

As the base of the marine food web, phytoplankton play a fundamental role in setting the productivity of the entire marine ecosystem. Specific phytoplankton groups also play key roles in the biogeochemical functioning of the ocean; for example, by fixing atmospheric nitrogen (diazotrophs) or silica cycling (diatoms). Additionally, the size structure of the community affects trophic interactions, food web productivity, and carbon sequestration potential[24–26]. Here, we explore how phytoplankton diversity responds to a high emissions climate change scenario, similar to RCP8.5[27,28], using a marine ecosystem model with 35 phytoplankton types and 16 zooplankton size classes[29–31], which are able to reorganise in response to changing oceanic conditions (see "Methods"). This model thus provides a more mechanistic representation of phytoplankton community structure than correlative or niche modelling approaches[32–34], and greater realism than Earth System Models (ESMs) used for IPCC projections[35,36].

Niche models and correlative approaches, by necessity, assume that the contemporary relationships between environmental conditions and phytoplankton abundance or diversity will remain the same in the future. These approaches do not have a mechanistic basis, and so changes in phytoplankton diversity driven by factors other than those included in the analysis (such as temperature, latitude, etc.), or conditions outside the bounds of variability in the contemporary ocean, cannot be reliably deduced. ESMs typically employ a very simplified ecosystem model, usually incorporating only 2 or 3 phytoplankton types. These models thus capture only a very limited diversity of phytoplankton communities. ESM results have focused on the response of phytoplankton to changing nutrient supply via changing stratification and circulation, which favours small species with high nutrient affinity[37,38]. However, in reality, phytoplankton respond to other factors which may result in changes to their relative competitiveness, or ultimately niche loss.

Here, we use a complex ecosystem model with multiple functional groups of phytoplankton and several size classes of both phytoplankton and zooplankton types. Diversity in the model is set by several different mechanisms: the ratio of the supply rate of different limiting nutrients, the supply rate of limiting nutrients, grazing pressure, and transport/mixing[39]. Previous analysis of the modelled diversity has demonstrated that the combination of limiting nutrient supply and grazing controls the number of size classes that co-exist, and the ratio of supply rates of limiting resources contributes to setting the number of co-existing functional groups[39]. Transport and mixing tend to increase local diversity[31]. Although this model incorporates considerably more complexity than climate models, nevertheless it can only capture a fraction of the huge diversity of phytoplankton in the real ocean. Specifically, we capture diversity within biogeochemical functional groups (for example, diatoms, diazotrophs, etc.) and size classes (Extended Data Fig. 1). However, we do not capture the diversity that arises due to other traits, such as thermal norms, morphology, or colony formation[39]. Thus, in this study, the terms 'richness' and 'diversity' reflect functional richness and diversity, and should be understood in the context of these two important trait axes within the many different axes that set biodiversity in the real ocean.

In this study, we quantify the response of marine phytoplankton diversity to climate change, focusing on future projections of community composition and turnover. We apply a high emissions climate scenario to a complex marine ecosystem model to explore the global and regional changes in phytoplankton community composition.

## Results

Phytoplankton biomass is projected to decrease over much of the tropical and subtropical ocean due to lower nutrient supply rate (Extended Data Fig. 2), consistent with previous studies[35,38,40,41]. Increases in phytoplankton biomass occur in high latitude regions due to the retreat of sea ice, longer growing seasons, and increased growth rates at higher temperatures, again consistent with previous work[36,37,40,41] (Fig. 1a). However, the increased ecological complexity of our model allows us to look beyond changes in biomass alone to uncover the community structure alterations that underlie the climate change response. Projected changes in biomass are in general reflected in phytoplankton richness, which declines by 2100 in large parts of the northern hemisphere subtropical and temperate regions (64% of area 23−55° N declines), and increases in polar and some equatorial regions (69% of area poleward of 55° or within 23° of the equator increases; Fig. 1b). In some tropical regions, up to 30% of modelled phytoplankton types become locally extinct, whereas in polar regions colonisation exceeds extinction, and richness increases by up to 30%.

The spatial patterns of phytoplankton functional group extinction and colonisation by the end of the century are illustrated in Fig. 2. Declining nutrient supply rates (Extended Data Fig. 2) drive the disappearance of less competitive and larger phytoplankton types[39] (indicated by the shallowing of the slope of the size spectrum; Fig. 1f), resulting in decreased richness in many northern hemisphere (sub)tropical regions (Fig. 2 and Extended Data Fig. 3). Lower nitrate relative to iron supply rate favours diazotrophs[42], and their range thus expands polewards, particularly in the northern hemisphere. In the same regions, diatom richness decreases with the reduction in silicic acid relative to nitrate[39] (Extended Data Fig. 2), resulting in the extinction of up to 30% of diatom types. Reduced nutrient supply and the

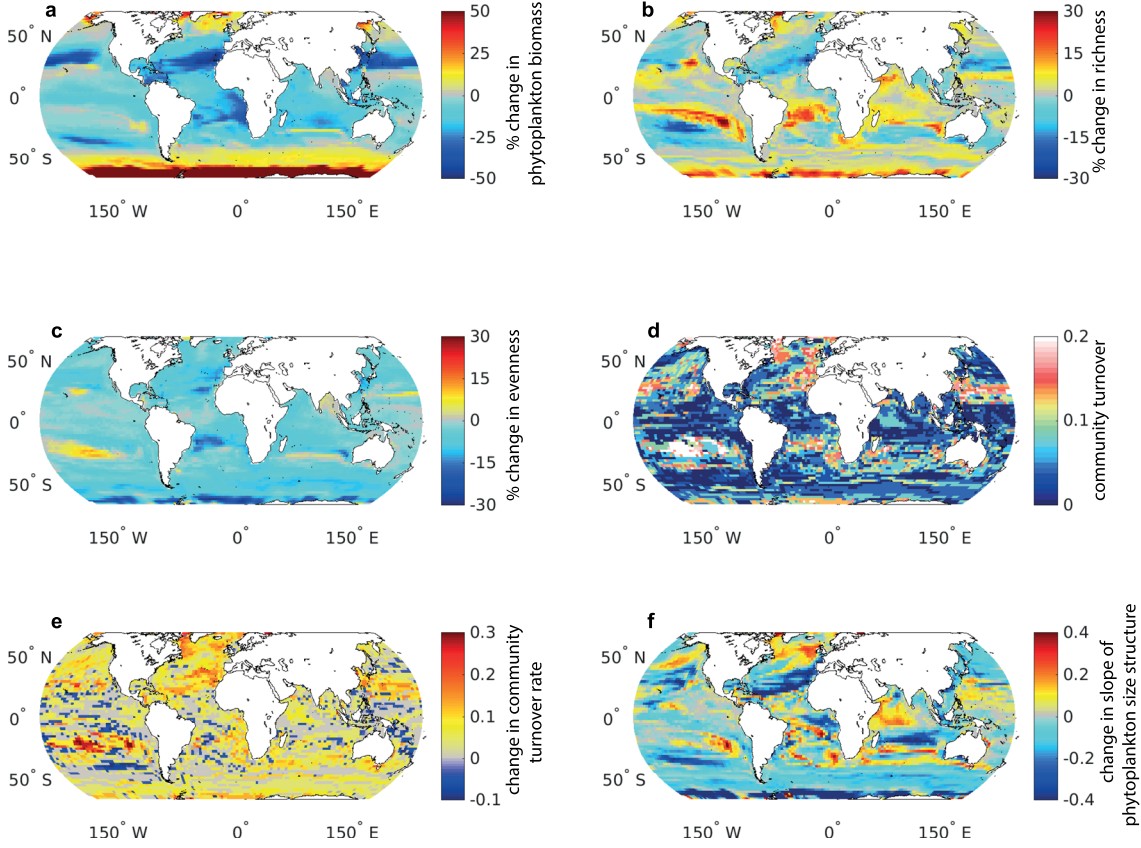

**Fig. 1 Change in phytoplankton community structure by the end of the century.** For subplots **a–d**, **f** the change between the baseline period (mean of 2005−2024) and end of the century (mean of 2081−2100) is shown. **a** Percent change in total phytoplankton biomass; **b** percent change in richness; **c** percent change in evenness; **d** community turnover; **e** change in community turnover rate (turnover between the mean of 2061−2080 and 2081−2100 minus mean of 2011−2030 and 2031−2040); **f** change in slope of the phytoplankton community size structure, where negative values indicate a greater abundance of small phytoplankton types.

subsequent loss of some larger zooplankton (Extended Data Fig. 4) results in fewer co-existing size classes[39]. Some mixotrophic dinoflagellate types become extinct by 2100 along subtropical gyre boundaries, but intermediate nutrient supply rate (Extended Data Fig. 2), and an increase in smaller phytoplankton (i.e., prey; Extended Data Fig. 4) allow them to become more competitive in, and ultimately colonise much of, the Southern Ocean. In contrast, the distribution of picoplankton, which are better adapted to low nutrient conditions, barely changes by 2100.

The Shannon diversity index, which incorporates both richness and evenness in biomass of co-existing types, declines almost globally (92% of ocean area; Extended Data Fig. 5f), driven primarily by an almost universal decrease in evenness (93% of ocean area; Fig. 1c). The declining evenness indicates that biomass becomes concentrated in fewer phytoplankton types by 2100 (Fig. 2; Extended Data Fig. 3).

Comparing the phytoplankton composition in the last decades of the century with the contemporary period (Fig. 1d) demonstrates that community turnover (i.e., the proportion of types that differ between two timepoints) is highest in parts of the temperate northern hemisphere and the South Pacific subtropical gyre, with ~20% of types being exchanged. Elsewhere, turnover is lower with <10% of phytoplankton types changing at the end of the century with respect to 2005−2024. The *rate* of turnover however increases in the majority of the ocean by the end of the century (63% of the area; Fig. 1e), indicating that phytoplankton community composition becomes increasingly variable (i.e., decreasingly stable) over time, both in regions of increasing and decreasing richness.

The slope of the phytoplankton size spectrum decreases by the end of the century in most sub-tropical regions (69% of the area) and in the Southern Ocean (90% of the area), i.e., the phytoplankton community shifts to dominance by smaller phytoplankton types (Fig. 1f). In the subtropics, the decrease in size spectrum is driven by a loss of relatively more large types than smaller types. On the other hand, the decrease in overall size of the community in the Southern Ocean is driven by a larger increase in smaller types than larger types (Extended Data Fig. 4). In some regions, there is an increase in overall size (33% of ocean area). In the case of the North Atlantic, this is driven by an influx of larger dinoflagellates and a general loss of diatoms (Fig. 2).

The results presented above should be interpreted within the limitations of the ecosystem model used which, although more complex than other global models, nevertheless only includes traits for functional group and size, but not for thermal norms. The modelled geographic shifts in plankton types are therefore not a direct response to warming temperatures (i.e., due to their thermal niches[40,43]), but instead are an indirect response occurring via changes to nutrient availability and relative competitiveness. However, the model metabolic processes (such as phytoplankton growth) do increase with warming waters, following an Arrhenius function[44]. Differences in temperature responses between types are likely to lead to alterations in their relative competitiveness, but such parameterisations are outside the scope of this study. The model also does not explicitly represent coastal regions (as the spatial resolution is too coarse), sea-ice communities are not explicitly modelled, and no anthropogenic impacts other than climate are simulated (e.g., runoff,

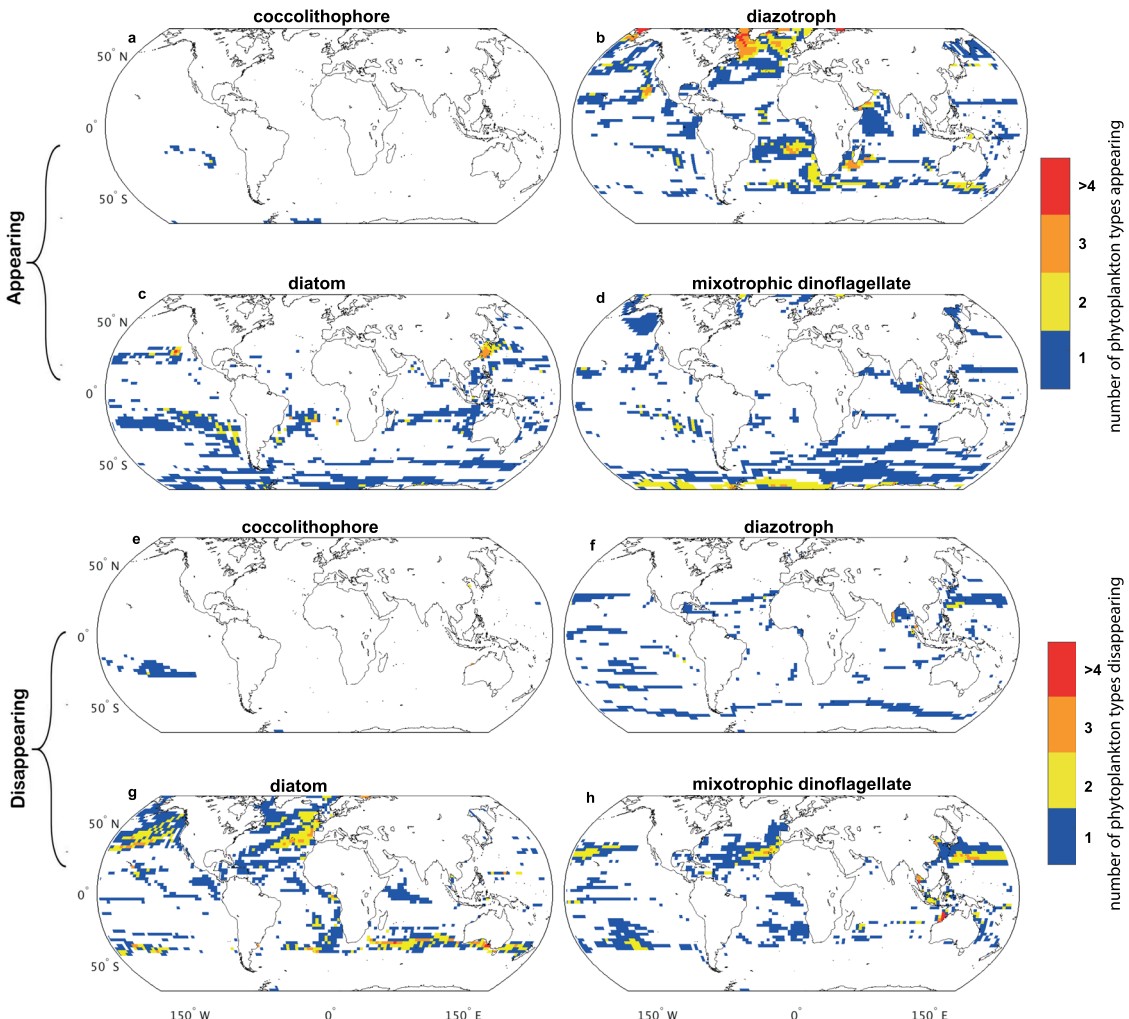

**Fig. 2 Appearance and disappearance of phytoplankton types by the end of the century.** The number of phytoplankton types appearing (**a–d**) and disappearing (**e–h**) between the baseline period (mean of 2005−2024) and end of century period (mean of 2081−2100) in each of six groups: coccolithophores (**a**, **e**), of which there are five types; diazotrophs (**b**, **f**), of which there are five types; diatoms (**c**, **g**), of which there are 11 types; mixotrophic dinoflagellates (**d**, **h**), of which there are 10 types. Prokaryotes and picoeukaryotes (of which there are two types of each) do not show any significant changes. Appearance (disappearance) is defined as a type contributing >0.1% (<0.1%) to total biomass.

pollution, habitat reduction). The modelled plankton also do not evolve or adapt to changing conditions; if plankton are able to do so on timescales comparable to those of climate change, then our results may represent a "worst-case scenario".

## Discussion

The complex ecological model used here provides insights into future changes in phytoplankton diversity and community turnover. As the model is mechanistic, rather than statistical (e.g., [33,34]), and represents significantly more complexity than typical ESMs (e.g., [36,38]), niche loss and changes in phytoplankton types' relative competitiveness can be assessed. Previous studies with variants of our model have examined the responses of different traits in historical runs, and knock-out and sensitivity experiments[39,40,43]. Here we present the analysis of a future climate change run, which focuses on changing phytoplankton diversity within the trait space of functional groups and size.

In the future ocean, our model predicts that biomass declines in the tropics and temperate regions (and increases at high latitudes), alongside significant shifts in phytoplankton community composition. In this model, changes in the supply of the limiting nutrient drive reduced biomass in lower latitudes[39], while

increased growth rates with warmer water lead to higher biomass in eutrophic high latitude regions. However multiple factors lead to the shifts in community structure. The reduction in macronutrient supply leads to declines in grazer abundance and trophic interactions; the combination can lead to a reduction in size classes (both lower richness and shallower slope of the size spectrum). Additionally, the ratio of supply of resources (nutrients and prey) affects the co-existence of functional groups[39]. For instance, previous analysis of this model has demonstrated that changes in the supply of nitrate relative to iron alters the distribution of diazotrophs, silica versus nitrate (or iron) supply alters diatom biogeography, and prey versus nutrient availability alters the mixotrophic dinoflagellate patterns[39]. Additionally increased stratification (and hence reduced mixing) contributes to altering diversity in the future ocean[39,45]. Together these mechanisms can lead to both positive and negative changes in richness (Fig. 1b).

Other studies have explored different aspects of phytoplankton diversity and response to future change using correlative approaches. The findings of a general decrease in diversity in the tropics and increase at high latitudes (particularly in the Southern Ocean), driven by colonisation exceeding extinction[46,47], broadly agree with our study (Figs. 1b, 2). The 'tropicalisation' of diversity

in temperate and polar latitudes[34] is also captured here as a shift towards smaller phytoplankton structure (Fig. 1f) and an increase in diazotrophs and mixotrophic dinoflagellates (Fig. 2). Discrepancies between the projections arise because the correlative approaches must assume that the modern-day relationship between phytoplankton species distribution and environmental conditions remains the same into the future, whereas we use a mechanistic model, which permits a dynamic response of phytoplankton diversity to changing environmental forcing.

Overall, we find a decline in Shannon diversity almost globally by the end of the century (Extended Data Fig. 5). However, this result masks an interesting interplay between richness and evenness ("Methods"). At temperate latitudes in the North Atlantic and North Pacific, the decline in Shannon index is driven by a decline in richness, implying that existing niches close in future conditions so that extinctions exceed colonisations (Fig. 1b, Extended Data Fig. 5). However, in the Southern Ocean, and off-equator in the Indian Ocean and South Atlantic, the declining Shannon index is associated with decreased evenness as richness increases (Fig. 1b, c). This indicates that, although colonisation exceeds extinction, the community becomes dominated by a few phytoplankton types, rather than its current more 'balanced' state.

In regions with high turnover and a decrease in richness (e.g., northwest North Atlantic, North Pacific gyre boundary; Fig. 1b, d), extinctions exceed colonisations, suggesting that the effect of climatic change is to reduce the number of potential niches[48]. Large scale changes in species composition occur due to environmental conditions exceeding the tolerances of phytoplankton types currently extant, so that they are outcompeted under future conditions by types adapted to lower nutrient supply rates, or whose co-existence relies on specific resource supply ratios. At the polar edges of our domain, an increase in richness coincides with high turnover, implying that expanding environmental niches lead to conditions favouring colonisation without excluding extant species[7]. This suggests that these regions may be refugia for phytoplankton types pushed beyond their tolerances at lower latitudes. The greater niche redundancy of some phytoplankton types, e.g., those with similar size replacements such as mixotrophs, may also make them less vulnerable to extinction.

Phytoplankton cell size has been called a "master trait" in ocean systems, as cell size ranges over 9 orders of magnitude[49]. In our study, the projected changes in phytoplankton size structure (Fig. 1f) imply an increasing dominance of smaller phytoplankton types. A trend toward smaller phytoplankton would have implications for both the oceans' ecological and biogeochemical function, as regions dominated by small phytoplankton typically support less productive food webs[50–52] and sequester less organic carbon in the deep ocean[26,53] than those dominated by larger size classes.

The striking increase in turnover rate by the end of the century in this high emission scenario (Fig. 1e) implies a reduction in niche diversity, resulting in an increased occurrence of ephemeral phytoplankton species, and fewer persistently dominant species. Higher trophic levels reliant on consistent availability of a few dominant phytoplankton types will need to adapt to a rapidly varying diet, which may be less palatable or nutritious. Increasing variability in community composition also implies a loss of ecological resilience[54,55], i.e., a reduced ability to maintain ecosystem function and structure under changing conditions[56]. Although turnover in contemporary phytoplankton communities can be high on a daily to seasonal timescale[57–59], species richness remains relatively stable[5,8]. Similarly, during environmental upheavals associated with glacial/inter-glacial periods, the diatom community structure largely recovered to its pre-perturbation structure on a ~1 million year time scale[60]. The return to an 'equilibrium ecosystem' state was associated with the ability of a

seed population of phytoplankton to retain resistance to environmental change, suggesting that low resistance to environmental change does not necessarily equate to community fragility. Whether the phytoplankton community could recover from the perturbation associated with anthropogenic climate change (which is uni-directional on multi-decadal time scales) remains an open question. We find here that under continuing climate change in a high emissions scenario, turnover increases with time, and functional richness changes become pronounced. This implies that phytoplankton community resilience evident in contemporary populations on interannual timescales[60–62] may be impaired by the end of the century, resulting in an increasingly unstable marine ecosystem.

Our results reveal the potential for significant future disruption to marine phytoplankton communities in response to climate change, particularly under continued high greenhouse gas emissions. The projections highlight the potential vulnerability of phytoplankton community structure to climate change by integrating the exposure to stressors and the community's sensitivity to those stressors. However, our model does not account for adaptation, which is likely to increase the ecological resilience of the phytoplankton community as tolerances shift to account for changing environmental conditions. Indeed, laboratory manipulation experiments have demonstrated that phytoplankton can adapt to new environmental conditions, such as warmer or more acidic waters, within a few hundred generations (i.e., 2−3 years[63,64]). However, the multiple mechanisms driving the changes discussed here (nutrient supply, nutrient supply ratios, grazer control, advection) are likely to be more difficult for phytoplankton to adapt to than modest changes in temperature[65]. If organisms cannot adapt sufficiently rapidly to the development of novel climatic conditions, community reorganisation, population collapse, or other abrupt ecological shifts, may occur[66,67]. However, the lack of adaptation in our model suggests that the results presented above may well be a 'worst case scenario'.

The potential for organisms to migrate in order to remain within analogous environmental conditions has been posited[68,69]. However, here we find that relocation of communities (Fig. 2), in terms of their size classes and functional groups, does not necessarily prevent extinction by 2100 (particularly at low latitudes), and of diatoms and larger phytoplankton globally. This implies that higher trophic levels may not only need to migrate to remain in analogous climatic conditions (e.g., by tracking isotherms[70]), but also to remain in regions of analogous diets. Although zooplankton migration speeds may be sufficiently rapid to keep pace with the northward movement of isotherms[71,72], some zooplankton groups are dependent on fatty acids from specific phytoplankton species to avoid starvation, complete their life cycle, and/or survive environmental stressors[73,74]. If zooplankton are either unable to acquire the necessary prey items, or regions of analogous climate and analogous diet do not overlap in future, significant changes to marine food webs are likely to occur.

Long-standing ecological theory posits that diversity loss results in ecosystem instability[9,75,76]. Here we demonstrate that climate change is likely to drive altered phytoplankton diversity, and in particular a reduction in evenness, resulting in wholesale reorganisation of phytoplankton communities, and increasing instability in community structure, which will present profound challenges to the productivity of the entire marine food web. Indeed, trophic amplification may result in greater changes at higher trophic levels of the marine food web than for phytoplankton[77]. Nations dependent on fishing for their main protein source, principally low to middle-income countries, are concentrated at low latitudes[78], where we predict substantial changes in phytoplankton diversity and biomass by the end of the century.

## Methods

The ecosystem model used here has been previously described[29,30] and is used in the configuration detailed in ref. [31]. The ecosystem model includes 35 phytoplankton and 16 zooplankton types in seven biogeochemical functional groups covering a size distribution from 0.6 to 2425 μm equivalent spherical diameter (Extended Data Fig. 1). The cycling of carbon, phosphorus, nitrogen, silica, iron, and oxygen is incorporated in the model. The plankton groups consist of 2 prokaryote, 2 pico-prokaryote, 5 coccolithophore, 5 diazotroph, 11 diatom, 10 mixotrophic dinoflagellate, and 16 zooplankton types. Note that mixotrophy is only considered to occur in the dinoflagellate group, and that limited observational data restricts the representation of mixotrophy in this (and all) ecological models. All groups are modelled with constant C:N:P:Fe stoichiometry using Monod kinetics. Parameters influencing phytoplankton growth, grazing, and sinking are size-related and differ between functional groups[39]. The maximum growth rates and grazing are also determined by phytoplankton cell size[39]; prokaryotes and picoeukaryotes (the smallest group) have the lowest nutrient affinity, while the fastest-growing types are in the 3 μm cell size range[79]. Zooplankton grazing uses a Holling III function; zooplankton will graze on plankton 5−15 times smaller than themselves, but prefer organisms 10 times smaller. Zooplankton are differentiated only by size and no differences in functionality are parameterised; we thus limit our analysis to diversity in the phytoplankton. This ecosystem model was chosen due to its high level of phytoplankton diversity, especially in terms of functionality. The model captures both size and biogeochemical differences between phytoplankton types, which impact both biogeochemical and foodweb dynamics. This ecosystem model has previously been shown to reproduce satellite and in situ observations of both size classes and functional types[30,31].

The physical framework is the MIT Integrated Global System Model (IGSM[80–82]). The ocean component has a 2° × 2.5° resolution in the horizontal, and 22 layers in the vertical, ranging from 10 m at the surface to 500 m thick at depth. The simulation is run from 1860 to 1990 with observed emissions of greenhouse gases, and from 1990 to 2100 with a high emissions scenario, similar to the Representative Concentration Pathway 8.5 (RCP8.5) used in the Coupled Model Intercomparison Project 5[83]. In this study, we focus on the period 2006−2100. The plankton distributions compare well with observations of both functional types and size distribution[84,85], as demonstrated in previous model validation work[29,30,39]. All analyses are performed on biomass integrated over the full ocean depth (to capture any deep biomass maxima). All analysis was performed in Matlab 2019a.

Any species that contribute less than 0.1% to total biomass at that location and timestep are excluded from further analysis. Functional richness is then defined as the number of phytoplankton types (biogeochemical functional types and size classes) that coexist at a particular location and timestep. The functional Shannon index is defined as:

$$\text{Shannon} = -\sum_{s=1}^{s} p_i \ln p_i \qquad (1)$$

where $s$ is the total number of phytoplankton types in a sample (i.e., richness), $i$ is the total biomass of individuals in one type and $p_i$ is the proportion of biomass in type $i$ relative to the total biomass across all types. The Shannon index decreases as both the richness and the evenness of the community decrease, where evenness is defined as: Shannon/ln(richness).

Turnover, which is the proportion of phytoplankton types that differ between two timepoints, is calculated as:

$$\text{turnover} = (N_G + N_L)/N_T \qquad (2)$$

where $N_G$ is the total number of phytoplankton types gained, $N_L$ is the total number of types lost, and $N_T$ is the total number of types observed in both timepoints. This metric captures the gross change in species composition and varies between 0 (all species persist) and 1 (all species change). Here we calculate the community turnover between the mean of the baseline period (2005−2024) and the mean of the end of the century (2081−2100), shown in Fig. 1d. The increase in community turnover rate is calculated as the turnover between the mean of 2061−2080 and 2081−2100 minus the mean of 2011−2030 and 2031−2040, shown in Fig. 1e.

The slope of the phytoplankton type size spectrum was calculated by summing the biomass within each of the 16 phytoplankton size classes. The slope of log(-biomass) against log(equivalent spherical diameter) was then estimated using a robust linear regression technique, the Theil−Sen estimator. The slope of the phytoplankton type size spectrum is then the slope of the regression plus 3, assuming purely spherical phytoplankton.

**Reporting summary**. Further information on research design is available in the Nature Research Reporting Summary linked to this article.

## Data availability

The model output is available from https://dataverse.harvard.edu/dataverse/gud-igsm, specifically, the depth-integrated biomass output used in this study is available from https://doi.org/10.7910/DVN/LWHQNS (ref. [86]).

## Code availability

The numerical model code is available from https://dataverse.harvard.edu/dataverse/gud-igsm, specifically at https://doi.org/10.7910/DVN/UA8VNU (ref. [87]).

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

## Acknowledgements

Support for S.A.H. and B.B.C. was received from NERC National Capability programme CLASS (Climate Linked Atlantic Sector Science), grant number NE/R015953/1, and funding from the European Union's Horizon 2020 research and innovation programme under grant agreement No 820989 (COMFORT). Support for S.R.A. was received from a

NERC PhD studentship, NE/1498876. S.D. acknowledges support from NASA grants NNX16AR47G and 80NSSC17K0561.

## Author contributions

S.A.H. and S.R.A. conceived the study, S.A.H. and B.B.C. undertook the analyses, S.D. generated the model output, all authors contributed to writing the paper.

## Competing interests

The authors declare no competing interests.
