## [Peer Review File · Nature Communications]

Reviewers' Comments:

Reviewer #1:

Remarks to the Author:

Henson et al use a ocean plankton model to assess the changes in aspects of phytoplankton diversity under a future climate scenario.

The manuscript is mostly well written and realitively easy to understand. However, some detailes on the calculation of compositional change needs to be improved. The conclusions seem somewhat sensationalized compared to the results.

I think a revised version of this manuscript can make a very valuable contribution reg projected changes in ocean biogdiversity & processes.

l. 196 f, Interpretation of data: "Here we demonstrate that climate change is likely to drive decreased phytoplankton richness" - this is in contrast to the actual changes in richness shown in Fig 1b, ext.Fig.3c, where changes are shown in both directions (+/-)

Diversity estimates. In Fig 1, Henson et al report richness together with Shannon as diversity metrics. As shannon is a combined metric of richness & evenness, showing shannon next to richness is not very useful; instead, show richness with evenness. In general, , as these two metrics indeed give complementary information on community diversity.

Temporal turnover is shown as proxy for community stability, which may have consequences on dynamics of higher trophic levels. However, from teh text it does not get clear over which time interval turnover is calculated. Does this include seasonal turnover? is it inter-annzual turnover? oplz clarify!

minor comments:

- phytoplankton functional groups - is mixotrophy considered only in dinoflagellates or also in other groups? pls specify

- l. 166 – to migrate IN ORDER to remain (add xx)

- l. 166-169 – revise sentence, structure unclear

- l. 170 ff (results)- this text on higher trophic level is not directly related to the results that are presented in the manuscript, it is rather an interpretation, so move to discussion

- extended data fig 1: which data is shown? Present? Future?

Reviewer #2:

Remarks to the Author:

This short paper deals with a very central question of present oceanography, i.e., how phytoplankton will respond to climate change and its predicted impacts on ocean life. Many studies in the last decades have addressed this question with various statistical or modeling approaches. Here a complex model is used based on 35+16 different taxa, while changes in diversity are explored based on two main phytoplankton traits: size and biogeochemical requirements. The final verdict is quite alarming because the model returns a scenario of lower biomass and biodiversity, higher instability and overall diminished productivity, and limited functioning of the marine ecosystem. Although potentially interesting, this paper is not clear in several points, which leaves room for doubts about the validity and novelty of the results presented.

While introducing many taxa certainly represents a considerable improvement, it is not clear why only nutrient availability is considered among the several different aspects and consequences of climate change. Other projections of diversity have shown the relevance of multiple parameters potentially affecting the distribution of taxa and the diversity of the phytoplankton communities, in some cases highlighting a lower effect of nutrients compared to other variables (e.g. Barton et al. 2016). In a recent paper by the same Authors, it is clearly acknowledged that 'it is unlikely that

any single or even combination of environmental variables will be able to explain patterns of diversity' (Dutkiewicz et al. 2020, Biogeosciences). The choice to focus on a single variable type should be motivated and clearly stated upfront in the title and abstract. In fact, while the general aim of the paper is clear, its specific questions should be expressed with more detail. In this respect, the introduction is far too short. It should be improved by clarifying the aims and framing them more extensively in the landscape of the many different papers addressing the same general question.

Similarly, the approach used to answer the specific questions should be described more extensively. The ecological model is quickly introduced in the last paragraph of the introduction as 'a novel configuration of a model' described elsewhere, with little justification for the choice is provided. The implications of the choice of a different physical model are not explained either. In another paper from the same Authors (Dutkiewicz et al. GBC 2020), the physical model has a higher resolution, but the quite informative Fig suppl. 1 here presented is basically the same as Fig. 1 in that paper.

It is hard to understand how the results of this study compare with those previously obtained with similar approaches and less taxonomic categories, and the extent to which the addition of complexity leads to a higher prediction power or a better understanding of mechanisms behind the changes predicted. As the Authors also admit, the complexity here is still enormously lower than that of the actual natural communities, which makes the use of the term 'richness' quite misleading, despite a disclaimer at the end of the introduction. While some results (e.g., the decrease in biomass) match those obtained in other studies, the decrease in diversity does not match the outcomes of other projections, e.g., those based on correlative approaches and metabarcoding data or with multiple stressors and envelope projection. Comparisons should be presented with recent papers that deal with similar topics and show different/similar results. For example, the turnover of taxa and their extinction should be discussed in the light of niche disappearance and redundancy discussed in other papers, some also including Authors of this paper. To summarise, the Authors should make an effort to highlight what is the added value of this paper against the background of similar projections from other studies and what new ecological insights were obtained through their analyses.

Other comments

The introduction is rather vague and fails in setting the appropriate scene for the study. With such a wealth of studies on the topic, the Authors should briefly present previous results obtained with other approaches and main ecological theories explaining them. The first paragraph presents quite broad concepts, while the second and third ones (lines 45-52 and 54-62) basically repeat the same concepts twice, that species will have to migrate, adapt or get extinct under the future scenario of human pressure and anthropogenic climate change. Besides choosing one of the two, the Authors should give more space to the indirect impacts of the climate stressors, through changes in water column structure, along with light and nutrient availability, the latter being the main factors considered in this paper.

Line 37. Wouldn't 'therefore' be more appropriate than 'However' in this case?

Line 70-71, please say a few more words on RCP8.5 and 'business as usual' scenarios, e.g. 'the most pessimistic IPCC trajectory for greenhouse gases concentration' or use 'severe ocean warming (RCP8.5)'

Line 86-88: The increase in biomass at high latitude is also 'consistent with previous studies' or at least with some of them. It reads as a novel outcome of this study.

Line 91. The blue color is not at all dominant in Fig. 1 b, which shows wide ocean areas with no changes or increased diversity, especially in the southern hemisphere. Not clear why the comment here, and in the conclusion of the paper, is that phytoplankton diversity will decline.

Although interesting, the results are only superficially backed up by ecological explanations for the patterns depicted. I am quite surprised that mixotrophs are so heavily impacted by changes in nutrients: they should be much less affected compared to other phytoplankton groups. Particularly phagotrophic species, presently proposed to be renamed as mixoplankton, should be able to overcome nutrient limitation by eating, while osmotrophs could also functionally enhance their capability to use DOM as a source of nutrients. Because of different thermal breadths among plankton taxa (Chen 2015, Journal of plankton Research), temperature should also influence future taxa distribution in a species- or group-dependent way, either reinforcing or compensating the effects of nutrient availability.

Line 143-144. I have not read the famous Baas-Becking paper, written in German, but the

paradigm is generally cited as a result of large population size and high dispersal capability of aquatic microbes, which would lead to low overall diversity and scarce spatial differences beyond those imposed by local environmental selection. I am not clear about the reason for this citation here, and how 'everything is everywhere' relates to the resistance of seed populations to environmental changes.

Line 149-150. Please provide some references for the evidence of the resilience of contemporary phytoplankton communities

Lines 249-251. The placement of this paragraph at the end of the methods is not appropriate. As written above, part of the concepts here should better go to the introduction, whereas another part would be more properly placed in the conclusions.

Reviewer #1 (Remarks to the Author):

Henson et al use a ocean plankton model to assess the changes in aspects of phytoplankton diversity under a future climate scenario.

The manuscript is mostly well written and realitively easy to understand. However, some details on the calculation of compositional change needs to be improved. The conclusions seem somewhat sensationalized compared to the results.

I think a revised version of this manuscript can make a very valuable contribution reg projected changes in ocean biogdiversity & processes.

Many thanks for your positive review.

l. 196 f, Interpretation of data: "Here we demonstrate that climate change is likely to drive decreased phytoplankton richness" - this is in contrast to the actual changes in richness shown in Fig 1b, ext.Fig.3c, where changes are shown in both directions (+/-)

This text has been changed to "drive altered phytoplankton richness".

Diversity estimates. In Fig 1, Henson et al report richness together with Shannon as diversity metrics. As shannon is a combined metric of richness & evenness, showing shannon next to richness is not very useful; instead, show richness with evenness. In general, , as these two metrics indeed give complementary information on community diversity.

Good point. Figure 1c (change in Shannon) has now been replaced with Extended Data Figure 3i (change in evenness).

Temporal turnover is shown as proxy for community stability, which may have consequences on dynamics of higher trophic levels. However, from teh text it does not get clear over which time interval turnover is calculated. Does this include seasonal turnover? is it inter-annzual turnover? oplz clarify!

For Figure 1d, the turnover is calculated between the mean of the baseline period (2005-2024) and mean of the end of century (2081-2100). In Figure 1e, the change in community turnover rate is calculated as the turnover between the mean of 2061-2080 and 2081-2100 minus the mean of 2011-2030 and 2031-2040. This information is provided in the caption of Figure 1. We have also now added it to the Methods section.

minor comments:

- phytoplankton functional groups - is mixotrophy considered only in dinoflagellates or also in other groups? pls specify

Yes, mixotrophy is only parameterized in dinoflagellates. We have added a note to this effect in the Methods section (line 310-311). There are certainly considerably more phytoplankton species that are mixotrophs in the real ocean, but here we have simplified to have just one functional group including mixotrophs. Ongoing work is looking at including more types of mixotrophs and mixotrophy, but is beyond the scope of this study.

- l. 166 – to migrate IN ORDER to remain (add xx)

Changed as suggested.

- l. 166-169 – revise sentence, structure unclear

We have revised this sentence (lines 280-284) and hopefully it is clearer now:

"The potential for organisms to migrate in order to remain within analogous environmental conditions has been posited. However, here we find that relocation of communities (Fig. 2),

in terms of their size classes and functional groups, does not necessarily prevent extinction by 2100 (particularly at low latitudes), and of diatoms and larger phytoplankton globally.”

- 1. 170 ff (results)– this text on higher trophic level is not directly related to the results that are presented in the manuscript, it is rather an interpretation, so move to discussion
We have moved this section to the Discussion.

- extended data fig 1: which data is shown? Present? Future?

Present. This information has been added to the figure caption. It is encouraging that it looks similar to one in Dutkiewicz et al, (GCB, not GBC) 2021 (not 2020). Although the model was run at a different spatial resolution, the physics and biogeochemistry are sufficiently similar to the model run used here that the resulting communities are similar.

Reviewer #2 (Remarks to the Author):

This short paper deals with a very central question of present oceanography, i.e., how phytoplankton will respond to climate change and its predicted impacts on ocean life. Many studies in the last decades have addressed this question with various statistical or modeling approaches. Here a complex model is used based on 35+16 different taxa, while changes in diversity are explored based on two main phytoplankton traits: size and biogeochemical requirements. The final verdict is quite alarming because the model returns a scenario of lower biomass and biodiversity, higher instability and overall diminished productivity, and limited functioning of the marine ecosystem. Although potentially interesting, this paper is not clear in several points, which leaves room for doubts about the validity and novelty of the results presented.

Thank you for the constructive review. We hope our revisions have clarified the presentation of our work, and emphasised the novelty of the results.

While introducing many taxa certainly represents a considerable improvement, it is not clear why only nutrient availability is considered among the several different aspects and consequences of climate change. Other projections of diversity have shown the relevance of multiple parameters potentially affecting the distribution of taxa and the diversity of the phytoplankton communities, in some cases highlighting a lower effect of nutrients compared to other variables (e.g. Barton et al. 2016). In a recent paper by the same Authors, it is clearly acknowledged that ‘it is unlikely that any single or even combination of environmental variables will be able to explain patterns of diversity’ (Dutkiewicz et al. 2020, Biogeosciences). The choice to focus on a single variable type should be motivated and clearly stated upfront in the title and abstract. In fact, while the general aim of the paper is clear, its specific questions should be expressed with more detail.

Diversity in this study (as described also in Dutkiewicz, 2020) is controlled by many different mechanisms. We find that the main driver of diversity is changes in the nutrient supply rate, rather than nutrient concentrations - as such we are not using a single environmental variable to explain everything. As well as the ratio of the supply rate of different limiting nutrients, the supply rate of the limiting nutrient, grazing pressure, and transport/mixing are all important in setting the diversity. In particular, the combination of supply of limiting nutrient and grazing sets the number of size classes that co-exist. The ratio of supply rates of limiting resources helps set the number of co-existing functional groups, particularly diazotrophs and diatoms. Transport and mixing tend to increase local diversity. We

acknowledge that we did not make this clear enough in the previous version of this paper, and have more strongly highlighted the different mechanisms at play in this revised version (see for instance lines 90-95). Lines 119-130 have also been rewritten to be more explicit that supply rates are the key term. Indeed, implicit (and more explicit in this revised version) in the discussion is that grazing pressures also alter. We have tried to clarify this by adding (lines 187-194): “The reduction in macronutrient supply leads to declines in grazer abundance and trophic interactions; the combination can lead to a reduction in size classes (both lower richness and shallower slope of the size spectrum). Additionally, the ratio of supply of resources (nutrients and prey) affects the co-existence of functional groups. For instance, changes in the supply of nitrate relative to iron alters the distribution of diazotrophs, silica versus nitrate (or iron) supply alters diatom biogeography, and prey versus nutrients alters the mixotrophic dinoflagellate patterns. Additionally increased stratification (and hence lower mixing) contributes to altering diversity in the future ocean (Dutkiewicz et al., 2020, Levy et al., 2015).”

In this respect, the introduction is far too short. It should be improved by clarifying the aims and framing them more extensively in the landscape of the many different papers addressing the same general question.

We have substantially rewritten the Introduction to provide more context for this study.

Similarly, the approach used to answer the specific questions should be described more extensively. The ecological model is quickly introduced in the last paragraph of the introduction as ‘a novel configuration of a model’ described elsewhere, with little justification for the choice is provided. The implications of the choice of a different physical model are not explained either. In another paper from the same Authors (Dutkiewicz et al. GBC 2020), the physical model has a higher resolution, but the quite informative Fig suppl. 1 here presented is basically the same as Fig. 1 in that paper.

The climate change simulation used here is only available on a coarser resolution grid due to computational constraints. It is encouraging that the two model resolutions produce similar biogeography, suggesting that the coarser resolution physics captures the important physical circulation and mixing. Dutkiewicz et al (GCB, 2021 – not GBC 2020) was in review when this paper was submitted and as such was not referenced. We have altered the text to now explain that we use the same ecosystem setup as in Dutkiewicz et al (2021), lines 305-316: “The ecosystem model used here has been previously described and is used in the configuration detailed in Dutkiewicz et al. (2021).... The ecosystem model was chosen due to its high level of diversity, especially in terms of functionality. The model captures both size and biogeochemical differences between plankton types, which impact both biogeochemical and foodweb dynamics. This ecosystem model has previously been shown to reproduce satellite and in situ observations of both size classes and functional types (Ward, 2015; Buitenhuis et al, 2013).”

It is hard to understand how the results of this study compare with those previously obtained with similar approaches and less taxonomic categories, and the extent to which the addition of complexity leads to a higher prediction power or a better understanding of mechanisms behind the changes predicted.

We have expanded both the Introduction and Discussion to address this point. It would be impossible to address diversity changes with a model with only 2 or 3 phytoplankton types (as the ESMs have), so the complexity of this model is essential for addressing both how diversity changes into the future, and the underlying mechanisms.

As the Authors also admit, the complexity here is still enormously lower than that of the actual natural communities, which makes the use of the term ‘richness’ quite misleading, despite a disclaimer at the end of the introduction. While some results (e.g., the decrease in biomass) match those obtained in other studies, the decrease in diversity does not match the outcomes of other projections, e.g., those based on correlative approaches and metabarcoding data or with multiple stressors and envelope projection. Comparisons should be presented with recent papers that deal with similar topics and show different/similar results. For example, the turnover of taxa and their extinction should be discussed in the light of niche disappearance and redundancy discussed in other papers, some also including Authors of this paper. To summarise, the Authors should make an effort to highlight what is the added value of this paper against the background of similar projections from other studies and what new ecological insights were obtained through their analyses.

We have significantly reworked both the Introduction and Discussion to address this point.

Other comments

The introduction is rather vague and fails in setting the appropriate scene for the study. With such a wealth of studies on the topic, the Authors should briefly present previous results obtained with other approaches and main ecological theories explaining them. The first paragraph presents quite broad concepts, while the second and third ones (lines 45-52 and 54-62) basically repeat the same concepts twice, that species will have to migrate, adapt or get extinct under the future scenario of human pressure and anthropogenic climate change.

The Introduction has been expanded to provide more context for this study and has also been restructured to avoid repetition.

Besides choosing one of the two, the Authors should give more space to the indirect impacts of the climate stressors, through changes in water column structure, along with light and nutrient availability, the latter being the main factors considered in this paper.

We’re unclear what the reviewer means by “choosing one of the two”, however, as described in our answer above, nutrient availability is not the only factor driving the diversity in this model. Hopefully the revised text makes this point more clearly.

Line 37. Wouldn’t ‘therefore’ be more appropriate than ‘However’ in this case?

Changed as suggested.

Line 70-71, please say a few more words on RCP8.5 and ‘business as usual’ scenarios, e.g. ‘the most pessimistic IPCC trajectory for greenhouse gases concentration’ or use ‘severe ocean warming (RCP8.5)’

On request from the editors, we have changed ‘business as usual scenario’ to ‘high emissions scenario’.

Line 86-88: The increase in biomass at high latitude is also ‘consistent with previous studies’ or at least with some of them. It reads as a novel outcome of this study.

We have altered the construction of this sentence to hopefully make clear that the increases at high latitude are also consistent with previous studies.

Line 91. The blue color is not at all dominant in Fig. 1 b, which shows wide ocean areas with no changes or increased diversity, especially in the southern hemisphere. Not clear why the

comment here, and in the conclusion of the paper, is that phytoplankton diversity will decline.

We changed the comment in the conclusion to highlight that diversity is altered (not necessarily declines). At line 91 in the original manuscript (now lines 111-117), we already highlighted both the regions of declining diversity and those of increasing diversity: “the increased ecological complexity of our model allows us to look beyond changes in biomass alone to uncover the community structure alterations that underlie the climate change response. Projected changes in biomass are reflected in phytoplankton richness, which declines by 2100 in most subtropical and temperate regions, and increases in polar and some equatorial regions (Fig. 1b). In many tropical regions, up to 30% of modelled phytoplankton types become locally extinct, whereas in polar regions colonisation exceeds extinction and richness increases by up to 30%.”

Although interesting, the results are only superficially backed up by ecological explanations for the patterns depicted. I am quite surprised that mixotrophs are so heavily impacted by changes in nutrients: they should be much less affected compared to other phytoplankton groups. Particularly phagotrophic species, presently proposed to be renamed as mixoplankton, should be able to overcome nutrient limitation by eating, while osmotrophs could also functionally enhance their capability to use DOM as a source of nutrients. Because of different thermal breadths among plankton taxa (Chen 2015, Journal of plankton Research), temperature should also influence future taxa distribution in a species- or group-dependent way, either reinforcing or compensating the effects of nutrient availability. *The mixotrophs in this model are phagotrophic, not osmotrophs, and we do not resolve different thermal breadths. Indeed, differences in temperature responses between taxa are likely to lead to some interesting alterations in their competitiveness, but such parameterizations are outside the scope of this study. We have added this point to line 164-165. The mixotrophs in the model compete best in regions where there are medium amounts of both nutrients and prey. As nutrient supplies reduce, they are favoured even more in what had been higher nutrient supply regions, which now have lower nutrients and also potentially more smaller phytoplankton (i.e. more prey). It is likely that the reduction in zooplankton will also provide a wider niche for them. We have added this point to line 125-128: “Some mixotrophic dinoflagellate types become extinct by 2100 along subtropical gyre boundaries, but intermediate nutrient concentrations, an increase in smaller phytoplankton (i.e. prey) and reductions in grazing pressure allow them to become more competitive in, and ultimately colonise much of, the Southern Ocean.”*

Line 143-144. I have not read the famous Baas-Becking paper, written in German, but the paradigm is generally cited as a result of large population size and high dispersal capability of aquatic microbes, which would lead to low overall diversity and scarce spatial differences beyond those imposed by local environmental selection. I am not clear about the reason for this citation here, and how ‘everything is everywhere’ relates to the resistance of seed populations to environmental changes.

We agree with the reviewer that this reference here is a bit confusing. We’ve removed it which hopefully improves the clarity of our message.

Line 149-150. Please provide some references for the evidence of the resilience of contemporary phytoplankton communities

We have added references here.

Lines 249-251. The placement of this paragraph at the end of the methods is not appropriate. As written above, part of the concepts here should better go to the introduction, whereas another part would be more properly placed in the conclusions.

This paragraph has now been incorporated into the Introduction and Results sections of the paper.

Reviewers' Comments:

Reviewer #1:

Remarks to the Author:

I mostly appreciate the revised version of the manuscript, Henson et al have addressed my (limited) concerns well.

Regarding the main conclusions, the term 'resilience' seems used in the wrong context. Resilience is defined as the ability of a system to return to the previous state following a disturbance. GC is a shifting baseline, the disturbance 'lasts'(Correctly stated in l 261!). However, Henson et al derive conclusions about resilience from the observation of increased temporal turnover as a response to the unidirectional change (shifting baseline) and not readily a signal of reduced resilience. To test for changes in resilience, the authors would need to analyse the resilience as a capacity of the community to return to the original state, both at present day as well as in the future scenario, under some pulse-disturbance (e.g. heat wave).

In terms of representation of phytoplankton diversity, the limited resolution of 'mixotrophs' is a downside of this paper. However, one must admit that the data available for parametrizing mixotrophic flagellates is still limited.

Reviewer #2:

Remarks to the Author:

The Authors have done a great job rearranging some parts of the manuscript, namely moving many comments and interpretations from results to discussion and also clarifying several obscure points of the paper by adding some of the explanations requested. The biological response seen through the model experiments is clear for several groups (but not for all groups, see below). The novelty of the model and the results compared to previous and similar studies has also been clarified. However, I still miss in this paper the demonstration of the link between the driving factors and the observed changes. Although in several parts of the paper – and in their rebuttal – the Authors have referred to nutrient supply, grazing and mixing conditions, there is a lack of clarity about what variables in the model are used as forcing and what instead are the results of the model. In their rebuttal, the Author state

We find that the main driver of diversity is changes in the nutrient supply rate, rather than nutrient concentrations - as such we are not using a single environmental variable to explain everything. As well as the ratio of the supply rate of different limiting nutrients, the supply rate of the limiting nutrient, grazing pressure, and transport/mixing are all important in setting the diversity.

My question here is: did the Authors run the model with different supply rates to come to this conclusion? Or, did they relate the results of the modeled diversity with nutrient supply rates? The results supporting their conclusion are not explained, nor the methods explain how this correlation/relationships between the observed change and environmental changes was analyzed. These aspects are probably obvious to people who are familiar with the model already used in other papers, but the readers should find here all useful information to fully understand the value of the results presented.

Although I am convinced that the paper represents a good contribution to our understanding of diversity in the future oceans, I think that some further effort should still be made to improve the description of the results, which are summarised in a quite dry and quick way. Besides the clarification about the role of nutrient supply as the driver of the observed changes, along with grazing pressure changes, some more comments would be interesting concerning the 16 zooplankton classes, their changes in diversity and the possible effects of those changes, while only results concerning the changes in size are presented.

The Authors refer to areas of the oceans affected by changes in diversity, abundance, size etc. in a descriptive way. Wouldn't it be better to quantify the extent of areas showing changes in terms of percentages? In some cases the maps are very small and it is difficult to retrieve this information. 'One of the two': I apologize for the lack of clarity here, I meant the choice of which sentence of the two expressing the same concept should have been left in the text.

I am still uncomfortable with the statement that 'much of' the temperate and subtropical zone will

see a decrease in diversity, especially considering the southern hemisphere. I suggest changing to 'large areas of...', particularly in the northern hemisphere

Line 119. Declining nutrient supply rates drive the disappearance...how this conclusion was reached?

Line 127, 'an increase in small phytoplankton (i.e. prey) and reductions in grazing pressure ':The former statement is not supported by fig. Extended data 2, but is shown in Fig 1 D, which should better be described before introducing Fig.2. As for the reduction in grazing pressure, if it is a result of the present model it should be shown or described somewhere up in the results, otherwise a citation is needed here.

Line 130. Data about picoplankton richness are not shown. Fig. extended data 2, cited in on line 121 in the context of richness changes at line 121, actually describes the contribution of those groups to biomass. The two aspects of the groups (richness and biomass) and their trends would need some more detailed description or more clear reference to the respective figures. The question here is whether richness and contribution to abundance decrease or increase in parallel, which does not find an obvious answer in the figures nor emerges from the text accompanying Fig.2 and Fig. extended data 2. The citation of the latter figure to support that the biomass becomes concentrated in fewer phytoplankton types by 2100 is also unclear.

Line 156. This sentence is misplaced in the results section, it should be moved to the discussion.

Line 159-162.'The modeled geographic shifts in plankton types are therefore not a direct response to warming temperatures (i.e. due to their thermal niches^{40,43}), but instead respond indirectly through changes to nutrient availability and relative competitiveness'. The second part of the sentence is not well written nor clear. Please rephrase,

Line 636. Delete 'all'. The whole legend should be rephrased, starting it with what is the figure about, i.e. the last sentence, and then the periods in the columns.

Reviewer #1 (Remarks to the Author):

I mostly appreciate the revised version of the manuscript, Henson et al have addressed my (limited) concerns well.

Regarding the main conclusions, the term 'resilience' seems used in the wrong context. Resilience is defined as the ability of a system to return to the previous state following a disturbance. GC is a shifting baseline, the disturbance 'lasts'(Correctly stated in l 261!). However, Henson et al derive conclusions about resilience from the observation of increased temporal turnover as a response to the unidirectional change (shifting baseline) and not readily a signal of reduced resilience. To test for changes in resilience, the authors would need to analyse the resilience as a capacity of the community to return to the original state, both at present day as well as in the future scenario, under some pulse-disturbance (e.g. heat wave).

There are 2 definitions of resilience commonly found in ecological literature:

- 1. The ability of an ecosystem to return to an equilibrium following a perturbation (Holling, 1973)*
- 2. The capacity of a system to absorb perturbations while undergoing change to maintain essentially the same functions, such as nutrient cycling or biomass production (Walker et al., 2004)*

The second is also called 'ecological resilience' (Gunderson, 2000). We are focusing on the 2nd definition here, so that our projected changes in diversity, biomass etc. are indicators of potential loss of resilience. To clarify this, we have changed the terminology to 'ecological resilience' in the text where appropriate and reworded lines 255-257.

In terms of representation of phytoplankton diversity, the limited resolution of 'mixotrophs' is a downside of this paper. However, one must admit that the data available for parameterizing mixotrophic flagellates is still limited.

Yes, we agree that the limited observational data restricts the parameterisation of mixotrophy in this (and all) ecological models. We have added a comment to this effect on lines 318-320.

References:

Holling, C. S. (1973), Resilience and Stability of Ecological Systems, Annual Review of Ecology and Systematics, 4(1), 1-23

Walker, B., C. S. Holling, S. R. Carpenter, and A. Kinzig (2004), Resilience, adaptability and transformability in social–ecological systems, Ecology and Society, 9(2), 5

Gunderson, L. H. (2000), Ecological Resilience – In Theory and Application, Annual Review of Ecology and Systematics, 31(1), 425-439

Reviewer #2 (Remarks to the Author):

The Authors have done a great job rearranging some parts of the manuscript, namely moving many comments and interpretations from results to discussion and also clarifying several obscure points of the paper by adding some of the explanations requested. The biological response seen through the model experiments is clear for several groups (but not for all groups, see below). The novelty of the model and the results compared to previous and similar studies has also been clarified.

Thank you for your positive and constructive comments.

However, I still miss in this paper the demonstration of the link between the driving factors and the observed changes. Although in several parts of the paper – and in their rebuttal - the Authors have referred to nutrient supply, grazing and mixing conditions, there is a lack of clarity about what variables in the model are used as forcing and what instead are the results of the model. In their rebuttal, the Author state:

“We find that the main driver of diversity is changes in the nutrient supply rate, rather than nutrient concentrations - as such we are not using a single environmental variable to explain everything. As well as the ratio of the supply rate of different limiting nutrients, the supply rate of the limiting nutrient, grazing pressure, and transport/mixing are all important in setting the diversity.”

My question here is: did the Authors run the model with different supply rates to come to this conclusion? Or, did they relate the results of the modeled diversity with nutrient supply rates? The results supporting their conclusion are not explained, nor the methods explain how this correlation/relationships between the observed change and environmental changes was analyzed. These aspects are probably obvious to people who are familiar with the model already used in other papers, but the readers should find here all useful information to fully understand the value of the results presented.

Apologies that we didn't make clear in our rebuttal that these conclusions had arisen from previously published work. Changes in nutrient supply occur over the 21st century as a result of changes in circulation and mixing (see new Extended Data Fig. 2). To reach these conclusions, we use insight from previous publications using this model which explored in detail the influence of differing supply rates, grazing pressure and advection on phytoplankton diversity (Dutkiewicz et al., 2013, 2014, 2020). Thus, given the climate change driven changes in nutrient supply, the phytoplankton community responded in a manner that we could explain given previously published work.

We have now added a new supplemental figure (Extended Data Fig. 2) showing the changes in nutrient supply and more carefully explained the results relative to this and other figures, as well as additional citations to show that these insights have come from previous work, e.g. on line 92-96 and line 198-201.

Although I am convinced that the paper represents a good contribution to our understanding of diversity in the future oceans, I think that some further effort should still be made to improve the description of the results, which are summarised in a quite dry and quick way. Besides the clarification about the role of nutrient supply as the driver of the observed changes, along with grazing pressure changes, some more comments would be interesting concerning the 16 zooplankton classes, their changes in diversity and the possible effects of those changes, while only results concerning the changes in size are presented.

We have substantially revised lines 122-137 to explain the results more thoroughly, and have also added an additional figure (Extended Data Fig. 2) and referred more to existing figures. The changes in the biomass of the 16 zooplankton are shown in Extended Data Fig. 4.

However, we consciously did not discuss the details of the changes in the 16 zooplankton classes and diversity, as zooplankton are only differentiated by size (and not function). They are relatively simply parameterized (relative to the phytoplankton) and are included in the model just to appropriately simulate the response of future phytoplankton biogeography and size spectrum. For this reason, changes to zooplankton are not the focus of either this modelling effort or of this study, and we instead focus here on the changes to phytoplankton diversity. We have added a comment to this effect on lines 320-322.

The Authors refer to areas of the oceans affected by changes in diversity, abundance, size etc. in a descriptive way. Wouldn't it be better to quantify the extent of areas showing changes in

terms of percentages? In some cases the maps are very small and it is difficult to retrieve this information.

The % area in which increases/decreases of some metrics occur was included in the original manuscript, but we have now expanded this reporting throughout the Results section.

‘One of the two’: I apologize for the lack of clarity here, I meant the choice of which sentence of the two expressing the same concept should have been left in the text.

Thank you for the clarification. Hopefully the revised text addressed this point.

I am still uncomfortable with the statement that ‘much of’ the temperate and subtropical zone will see a decrease in diversity, especially considering the southern hemisphere. I suggest changing to ‘large areas of...’, particularly in the northern hemisphere

We have changed the wording here as suggested and included % area in which increases/decreases occur to further quantify our statement (line 115-118).

Line 119. Declining nutrient supply rates drive the disappearance...how this conclusion was reached?

We show the declining nutrient supply in the new Extended Data Fig. 2. The insight into why reduced nutrients leads to fewer large species comes from Dutkiewicz et al. (2020), and we now reference this paper in this sentence as well as direct the reader to the new figure.

Line 127, ‘an increase in small phytoplankton (i.e. prey) and reductions in grazing pressure’: The former statement is not supported by fig. Extended data 2, but is shown in Fig 1 D, which should better be described before introducing Fig.2. As for the reduction in grazing pressure, if it is a result of the present model it should be shown or described somewhere up in the results, otherwise a citation is needed here.

We agree that the increase in smaller phytoplankton (though real) is not easily seen in Extended Data Fig. 3, but it is clearer in Extended Data Fig. 4 – which we now reference. We have removed the statement about reduction in grazing pressure, as this is in fact not germane to the argument. We do however now also reference the new Extended Data Fig. 2.

Line 130. Data about picoplankton richness are not shown. Fig. extended data 2, cited in on line 121 in the context of richness changes at line 121, actually describes the contribution of those groups to biomass. The two aspects of the groups (richness and biomass) and their trends would need some more detailed description or more clear reference to the respective figures. The question here is whether richness and contribution to abundance decrease or increase in parallel, which does not find an obvious answer in the figures nor emerges from the text accompanying Fig.2 and Fig. extended data 2. The citation of the latter figure to support that the biomass becomes concentrated in fewer phytoplankton types by 2100 is also unclear.

We have changed the sentence on picoplankton to read “In contrast, the distribution of picoplankton, which are better adapted to low nutrient conditions, barely changes by 2100.” (line 135-137).

We’re unclear on the relevance of the reviewer’s question of “whether richness and contribution to abundance decrease or increase in parallel” here. Richness and abundance are not necessarily related, as it’s possible to have fewer species (decreased richness), but at the same time either greater or lesser abundance of those species (and similarly for increased richness). The discussion in the section highlighted by the reviewer is intended to highlight the regions where the (dis)appearance of specific phytoplankton types occurs (i.e. describing, principally, Figure 2).

The citation to support that “the biomass becomes concentrated in fewer phytoplankton types by 2100” should have referred to Extended Data Fig. 3 – apologies for the confusion.

Line 156. This sentence is misplaced in the results section, it should be moved to the discussion.

The line numbering seems to be slightly off in the PDF and Word versions of our manuscript, but we think the reviewer is referring to the sentence “In the case of the North Atlantic, this is driven by an influx of larger dinoflagellates and a general loss of diatoms due to increase silica limitation.” We have amended this slightly to “In the case of the North Atlantic, this is driven by an influx of larger dinoflagellates and a general loss of diatoms (Fig. 2).” (line 161-163)

Line 159-162. ‘The modeled geographic shifts in plankton types are therefore not a direct response to warming temperatures (i.e. due to their thermal niches^{40,43}), but instead respond indirectly through changes to nutrient availability and relative competitiveness’. The second part of the sentence is not well written nor clear. Please rephrase,

We have rewritten this sentence to read (line 167-170): “The modelled geographic shifts in plankton types are therefore not a direct response to warming temperatures (i.e. due to their thermal niches), but instead are an indirect response occurring via changes to nutrient availability and relative competitiveness.”

Line 636. Delete ‘all’. The whole legend should be rephrased, starting it with what is the figure about, i.e. the last sentence, and then the periods in the columns.

We have changed the caption on Extended Data Fig. 5 to read (line 656-659):

“Phytoplankton richness (a-c), Shannon index (d-f), and evenness of phytoplankton population (g-i) for mean of baseline period (2005-2024) in left column, mean of end of century period (2081-2100) in centre column, and difference between the end of century and baseline periods in right column.”

Reviewers' Comments:

Reviewer #2:

Remarks to the Author:

The Authors have exhaustively addressed many of my concerns and improved the 'Results' section adding clarity and accuracy to several comments to the data. I am still somewhat unclear on the conclusion that nutrient supply rates will drive the future changes of biodiversity but now the Authors have clarified that the relevant information is in previous papers and have added citations. In case some Figures added to the Extended data belong to previous studies, namely ext. Figs 1 and 2, those studies should be mentioned in the legends.

REVIEWERS' COMMENTS

Reviewer #2 (Remarks to the Author):

The Authors have exhaustively addressed many of my concerns and improved the 'Results' section adding clarity and accuracy to several comments to the data. I am still somewhat unclear on the conclusion that nutrient supply rates will drive the future changes of biodiversity but now the Authors have clarified that the relevant information is in previous papers and have added citations. In case some Figures added to the Extended data belong to previous studies, namely ext. Figs 1 and 2, those studies should be mentioned in the legends.

Thanks for your positive comments. The figures mentioned have been drawn specifically for this paper, although the model output used to construct them was described in a previous study.